# Design of Advanced Human–Robot Collaborative Cells for Personalized Human–Robot Collaborations

Alessandro Umbrico [1], Andrea Orlandini [1], Amedeo Cesta [1], Marco Faroni [2,*], Manuel Beschi [2], Nicola Pedrocchi [2,3], Andrea Scala [4], Piervincenzo Tavormina [4], Spyros Koukas [5], Andreas Zalonis [5], Nikos Fourtakas [6], Panagiotis Stylianos Kotsaris [6], Dionisis Andronas [6] and Sotiris Makris [6]

1 National Research Council of Italy, Institute of Cognitive Sciences and Technologies, 00185 Rome, Italy; alessandro.umbrico@istc.cnr.it (A.U.); andrea.orlandini@istc.cnr.it (A.O.); amedeo.cesta@istc.cnr.it (A.C.)
2 National Research Council of Italy, Institute of Intelligent Industrial Technologies and Systems for Advanced Manufacturing, 20133 Milan, Italy; manuel.beschi@unibs.it (M.B.); nicola.pedrocchi@stiima.cnr.it (N.P.)
3 Dipartimento di Ingegneria Meccanica e Industriale, University of Brescia, 25123 Brescia, Italy
4 CEMBRE S.p.A., 25135 Brescia, Italy; andrea.scala@cembre.com (A.S.); piervincenzo.tavormina@cembre.com (P.T.)
5 Netcompany-Intrasoft, L1253 Luxembourg, Luxembourg; spyros.koukas@intrasoft-intl.com (S.K.); andreas.zalonis@netcompany-intrasoft.com (A.Z.)
6 Laboratory for Manufacturing Systems & Automation, University of Patras, 26500 Patras, Greece; nfourtakas@lms.mech.upatras.gr (N.F.); kotsaris@lms.mech.upatras.gr (P.S.K.); andronas@lms.mech.upatras.gr (D.A.); makris@lms.mech.upatras.gr (S.M.)
* Correspondence: marco.faroni@stiima.cnr.it

**Abstract:** Industry 4.0 is pushing forward the need for symbiotic interactions between physical and virtual entities of production environments to realize increasingly flexible and customizable production processes. This holds especially for human–robot collaboration in manufacturing, which needs continuous interaction between humans and robots. The coexistence of human and autonomous robotic agents raises several methodological and technological challenges for the design of effective, safe, and reliable control paradigms. This work proposes the integration of novel technologies from Artificial Intelligence, Control and Augmented Reality to enhance the *flexibility* and *adaptability* of collaborative systems. We present the basis to advance the classical *human-aware* control paradigm in favor of a *user-aware* control paradigm and thus personalize and adapt the synthesis and execution of collaborative processes following a *user-centric* approach. We leverage a manufacturing case study to show a possible deployment of the proposed framework in a real-world industrial scenario.

**Keywords:** human–robot collaboration; augmented reality; cyber physical systems; knowledge representation; planning and scheduling

## 1. Introduction

Human–robot collaboration (HRC) is expected to be a core element of future factories. Combining the repeatability and tirelessness of robots with humans' versatility and problem-solving skills often boosts the flexibility and productivity of industrial processes. However, the design of effective methodologies to steer the deployment of this new paradigm in real production environments is an open challenge for both researchers and companies [1]. According to [2], work organization and technical solutions for Cyber–Physical Systems (CPS) are supposed to evolve between two extreme alternatives: (i) the *techno-centric scenario* and; (ii) the *anthropo-centric scenario*. In the techno-centric scenario, the technological aspects dominate the organization of the work. In contrast, in the anthropo-centric scenario, human workers control the work, and technology helps them make decisions.

Existing approaches to CPS (and, among them, HRC) oscillate between these two extremes. However, human factors have gained more attention in the design of novel

methodologies for personalized production dynamics based on the operators' preferences, technical skills, and health-related issues.

Regarding HRC, human factors have been considered at different levels [3]. For example, optimization of human factors have been embedded into a task scheduler [4]; task allocation have been exploited to reduce the workload of human workers [5,6]; task synergy between human–robot tasks were optimized to reduce the cycle time [7]; human-aware motion planners demonstrated to be preferable by human users [8]. The works mentioned above mainly focus on the planning aspects of HRC. However, they disregard the complex effect of human–robot communication on the user experience. A stuttering human–system communication is often a major bottleneck to a fruitful collaborative process. For this reason, the communication between the human operator and the system is an object of intense study. In this regard, Augmented Reality (AR) is a striking tool able to overlay instructions and knowledge from CPSs to the physical operator's view [9].

Driven by this consideration, an EU-funded research project called SHAREWORK (EU Horizon 2020—https://sharework-project.eu (accessed on 21 June 2022)) aims to provide an all-around approach to HRC, where robots and operators collaborate at different cognitive and physical levels. A key objective of SHAREWORK is to make implicit and explicit communications between robots and humans smooth and fruitful. Explicit communications leverage multi-modal technologies and, in particular, Augmented Reality tools. Implicit communications require the robotic system to reason on the operator's intentions and act consequently. Therefore, task representation and planning are fundamental to provide the robot with the necessary autonomy and suitable initiative.

This paper presents some of the results of the SHAREWORK project. In particular, it presents the design methodology and deployment actions needed to provide a *user-aware* approach to HRC that enhances the flexibility of HRC systems. In addition, human-aware paradigms usually consider a one-fits-all solution, considering the human an anonymous agent. Here, we go beyond this concept and propose a user-centric methodology to shape the robots' behavior based on the specific characteristics of a single user (e.g., age, skills, experience) and preferences (e.g., left-handed vs. right-handed) [10], i.e., implementing personalized robot behavior that can better serve the human operator and, potentially, increase the technology perception and acceptance. Therefore, we propose the integration of planning, perception and communication into a unified technological framework.

An AI-based Knowledge Representation and Reasoning module encapsulates a user model representing features of human workers that are relevant with respect to production needs (e.g., match users' skills to the requirements of production tasks). Combined *AI-based task* and *motion planning* modules reason on this knowledge to coordinate human and robot agents taking into account known skills and features of the worker, while pursuing an optimization perspective. Furthermore, an AR-based *human–system interaction module* realizes advanced interaction mechanisms to contextualize communication to and from the worker to facilitate explicit human–robot communications and collaboration.

The paper is structured as follows: Section 2 illustrates aims and objectives of the SHAREWORK project, within which the methodology was developed; Section 3 discusses the user models and the knowledge-based formalism to represent users and production information; Section 4 shows how the framework embeds user-awareness, with a particular focos on the planning and communication modules; Section 5 discusses the integration of the proposed framework into a manufacturing scenario.

## 2. The SHAREWORK Project

The SHAREWORK project developed a safe and effective control system for anthropocentric HRC in fence-less environments. The project's developments follow the SHAREWORK architecture, a modular, distributed, service-oriented architecture (SoA) that defines a set of 15 different software and hardware modules designed as stand-alone, interacting components communicating through well-defined interfaces. The architecture has been designed to be fully interoperable and support various module configurations that can be

customized according to industrial needs. The SHAREWORK project architecture includes modules that understand the environment and human actions through knowledge and sensors, predict future state conditions, implement smart data processing, provide augmented reality and gesture and speech recognition technology.

### 2.1. General Architecture and Module Overview

Figure 1 shows a high-level overview of the SHAREWORK architecture, depicting the set of different modules and the high-level flow of information among them. The picture highlights the interconnection of Workspace Cognition, Planning, and human–robot communication composing the backbone of the architecture. Notwithstanding the modularity of the proposed approach, the core modules are combined into a user-centric framework oriented to user preferences and human factors at all levels (e.g., process representation, robot motion, human communication). Not only the single modules are user-centric *per se*, they are connected in such a way that the output of each module contain helpful information to enhance the user-awareness of other modules. It is the case, for example, of the trajectories planned by the Action and Motion Planner, which are visualized by the Human-System Communication module, or the user profiles stored in the Knowledge Base, which instantiates different communication interfaces based on the users' preferences.

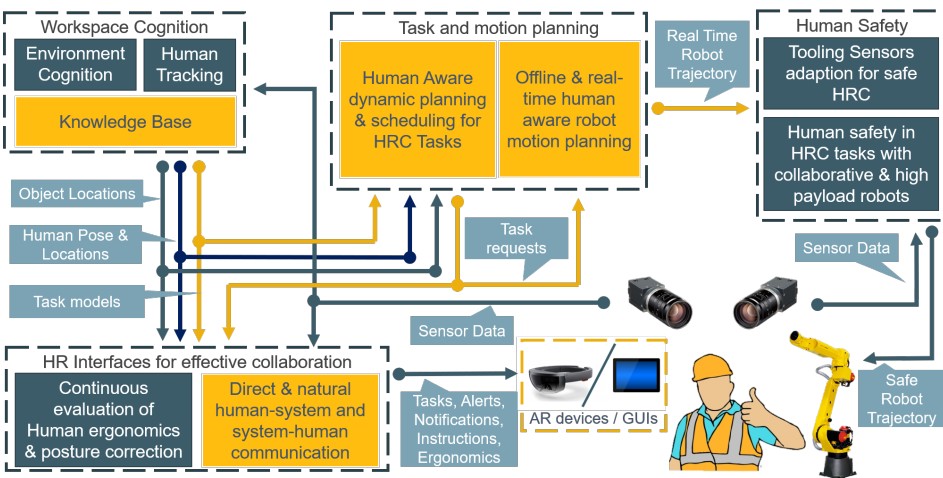

**Figure 1.** Overview of the SHAREWORK architecture.

In the following, we specifically focus on how each of the main modules support *user-awareness* within the proposed framework:

- **The Knowledge Base Module** stores a formal representation of the current status of the production environment based on the SHAREWORK ontology [11]. This module aggregates and elaborate information gathered from other modules to infer contextualized knowledge concerning for example situations/states of a worker, of the environment, of a production process being executed.
- **The Task Planning Module** coordinates the worker and the robot to cooperatively carry out production processes. In particular, this module synthesize a flexible temporal schedule of the tasks the worker and the robot should perform by taking into account *uncertainty*, *safety* and *efficiency*.
- **The Action and Motion Planning Module** receives a task from the Task Planning Module and finds a sequence of feasible movements to execute it. It comprises an Action Decomposition layer that converts a high-level task (e.g., pick an object, screw a bolt) into a sequence of motion planning problems. Then, it uses a motion planning algorithm to solve each problem and returns a sequence of trajectories that executes the high-level task. To consider the user, the Action and Motion Planning Module runs online; that is, all trajectories are calculated on the fly, just before their execution. To do so, it exploits human tracking data, usually acquired through a vision system.

This is necessary for two reasons: first, avoiding collisions and interference with the user (who is moving in the cell); second, adapting to changes in the environment (e.g., the user may move objects and tools during the work).

- **The Human-System Interaction Module** provides a bidirectional communication framework between operators and the SHAREWORK system. By incorporating various interface devices and sensors, a multi-modal interaction pipeline is structured to facilitate communication of (i) data and goals to the system (by the user) and; (ii) pending and current tasks, robot trajectories, event notifications, report results to the operator (by the system). Communication channels include AR devices and tablet interfaces. Supported by the knowledge base's ontology, the human system interaction module can be tailored to the operator's preferences and needs to establish an intuitive and user-aware working environment.

### 2.2. General Integration of Modules Supporting Personalized Collaboration

This section discusses how the modules introduced above work together for *user-awareness*. Figure 2 shows the integration of these modules and the information and control flow. Communication mechanisms and the exchange of messages/signals among the modules rely on ROS (https://www.ros.org (accessed on 21 June 2022)). Each module indeed defines a set of ROS topic and ROS services used to offer information/functionalities to and gather the necessary information from other modules.

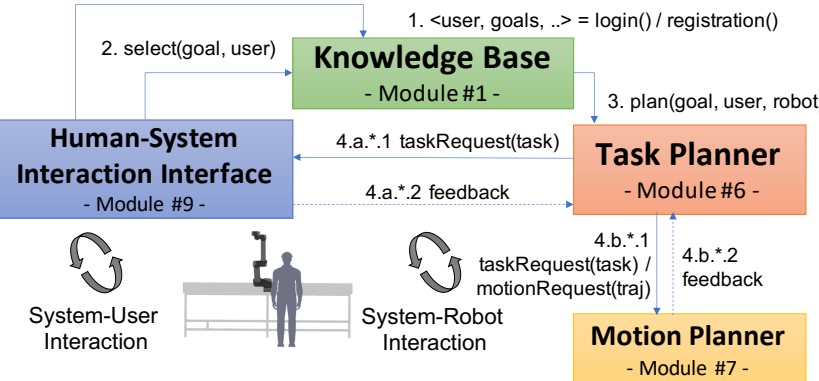

**Figure 2.** Integration schema of the considered modules. First a user logs into the system through the "Human-System Interaction Interface" and selects a production goal to perform (step 1 and 2). The "Task Planner" module receives a planning request (step 3) and synthesizes a contextualized collaborative plan to execute (step 4). Plan execution entails the simultaneous dispatching of multiple requests (see the "*.1" tags of step 4) to the "Human-System Interaction Interface" and to the "Motion Planner" asking the execution of planned tasks to the human and to the robot respectively. Every time a task is dispatched to the robot or to the human the planner receives a feedback about its execution (see the "*.2" tags of step 4) and adapts the plan to the observed state of the environment if necessary (replanning).

First of all, the Human-System Interaction Interface authenticates a particular worker into the system and retrieves information about his/her *user profile* (e.g., data from previous sessions, preferences, known skills) and information about the production context (e.g., known production goals, related production procedures, skills of the collaborative robot). The worker decides the production goal to perform and sends a "starting signal" to the Knowledge Base module through the Human-System Interaction Interface. The generated message specifies the production goal and the ID of the user that takes part to the process.

The Knowledge Base receives this signal through a subscribed ROS topic, contextualizes knowledge (e.g., infer the subset of operations the worker and the robot can actually perform) and configures the Task Planner by calling a dedicated ROS service.

This service specifically allows the Knowledge Base to automatically define the *control variables* of the planning model according to the requested production goal and the profile of the user (e.g., robot capabilities, operator skills, performance profile). The Task Planner then synthesizes and executes an optimized task plan. During the execution, the module dispatches task requests to the Human-System Interaction Interface and the Action and Motion Planner to interact with the robot.

The Human-System Interaction Interface displays information on the tasks requested to the human and waits for feedback from the operator. This ensures the correct dispatching of the task plan to the human actor. Similarly, the Action and Motion Planner receives tasks' requests for the robot and puts them into action. After the execution of the task, it sends feedback to the Task Planner to inform it about the outcome. The Human-System Interaction Interface and the Action and Motion Planner offer a set of ROS Actions that enable visualization and monitoring of human and robot tasks. For example, the Action and Motion Planner informs the Human-System Interaction Interface on the future robot trajectories so that they can be visualized on an interface (e.g., through AR).

### 2.3. Modular Deployment through Containerization

The modularity of the SHAREWORK architecture is also reflected in the software packaging. The system is a ROS-based system. It uses the Docker toolset to increase productivity, reduce the setup time in complex environments, and easily configure a customized version of the SHAREWORK architecture. In particular, each module is packaged in a separate Docker image and uploaded to a docker repository. An exception to this rule is the software modules with specific run-time requirements (e.g., the Android application of the *Human-System Interaction module* running on an Android tablet). By using the Docker compose tool that enables the definition and execution of multi-container applications, the SHAREWORK system can be configured to run in different configurations using an appropriate configuration file. Then, a SHAREWORK instance can be created and started with a single command.

## 3. Ontology-Based Model of Workers

HRC scenarios pursues a tight "teamwork" between the human and the robot requiring shared view and "mutual understanding" of the objective, constraints, capabilities and limitation of each other member as well as an implicit or explicit agreement about the procedure to follow [12–14]. The ontological model supports the effective coordination of human and robotic agents by providing a formal representation of: (i) production objectives, tasks and operational constraints; (ii) worker and robot capabilities/skills; (iii) known performances, preferences and physical/behavioral features of workers that may affect the interactions with the robot and the resulting collaborative processes.

### 3.1. Context-Based Ontology for Collaborative Scenarios

The SHAREWORK Ontology for Human–Robot Collaboration (SOHO) has been introduced in [11] as a general model characterizing collaborative dynamics between human and robot agents acting in a manufacturing scenario. It therefore defines the formal model (TBox) the *Knowledge Base Module* uses to build an abstraction of the production environment (ABox) and infer/contextualized useful information. SOHO is organized into a number of *contexts*, each defining concepts and properties that characterize an HRC scenario with respect to a particular perspective. A knowledge base is structured in shape of Knowledge Graphs (KGs) [15,16] and thus can be manipulated through standard semantic technologies based on OWL [17]. Specifically, the Knowledge Base Module of SHAREWORK has been developed using the open-source software library Apache Jena (https://jena.apache.org (accessed on 21 June 2022)).

As shown in [11] the environment, behavior and production contexts describe respectively: (i) physical entities and observable properties of a collaborative environment; (ii) skills and capabilities of the human and robot; (iii) production goals, tasks, and con-

straints of the HRC process. The behavior context uses the concept of `Function` [18] to correlate production tasks with the low-level operations the worker and the robot can actually perform (i.e., the functions).

### 3.2. Functions and Production Requirements

The goal of SOHO is to characterize production objectives, human and robot capabilities and thus contextualize operations they can perform to carry out production tasks collaboratively. The concepts `Cobot` and `HumanWorker` are defined as a specialization of the `DUL:Agent`. The acting qualities of each agent are represented by means of `Capability` and `Function`. Capabilities characterize competencies that agents have according to their structures and skills. For example, a human worker can perform welding operations if she is skilled in that task. Similarly, a robot can perform "pick and place" of objects if it is endowed with a gripper. Figure 3 shows an excerpt of SOHO pointing out the taxonomical structure of the concepts representing different types of production tasks.

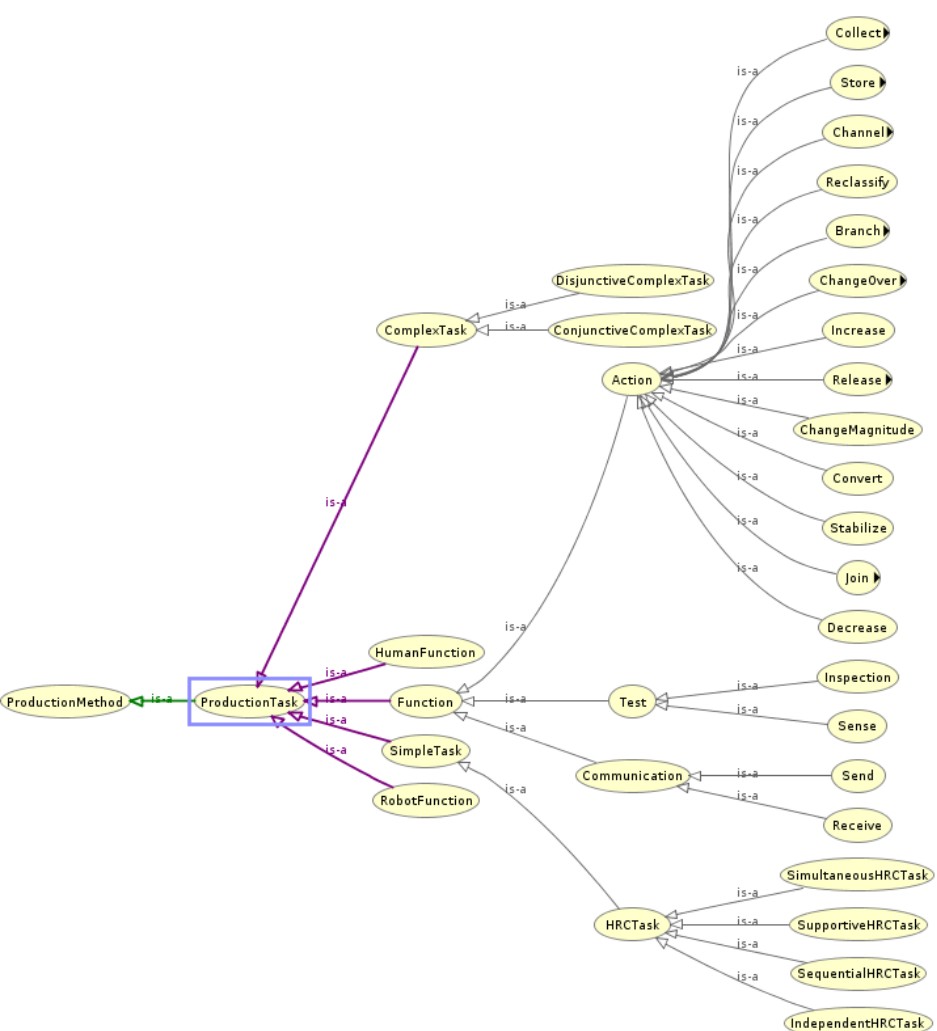

**Figure 3.** Excerpt of SOHO showing the taxonomical structure of production tasks. In particular the picture shows the integrated taxonomy of functions introduced in [18].

While capabilities do not depend on the features of a production context, the concept of `Function` characterizes low-level production tasks humans and robots should perform in a manufacturing environment. SOHO integrates the Taxonomy of Functions defined in [18] and defines different types of `Function` according to the effects they have on `DUL:Quality` of objects. The instances of `Function` a generic agent can perform can be dynamically inferred according to actual capabilities of that agent. Namely, the model of

Function proposed in [18] is extended to correlate them to the set of Capability needed
to correctly perform them. The separation between functions and capabilities supports
contextual reasoning since functions contextualize general agents' capabilities with respect
to the needs of a production scenario.

$$
\begin{aligned}
\texttt{Function} \sqsubseteq \quad & \texttt{ProductionTask} \sqcap \\
& \exists\,\texttt{DUL:isDescribedBy.ProductionNorm} \sqcap \\
& \exists\,\texttt{canBePerdformedBy.DUL:Agent} \sqcap \\
& \exists\,\texttt{hasEffectOn.DUL:Quality} \sqcap \\
& \exists\,\texttt{hasTarget.ProductionObject} \sqcap \\
& \exists\,\texttt{requires.ProductionObject} \sqcap \\
& \exists\,\texttt{requires.Capability}
\end{aligned}
\tag{1}
$$

The description of a production process follows a task-oriented approach. The top-
level element is the ProductionGoal which defines the general objectives of a produc-
tion context. Each ProductionGoal is associated with a number of ProductionMethod
(at least one method for each goal is necessary) specifying production and operational
constraints. Each ProductionMethod always refers to one ProductionGoal and is com-
posed by a hierarchical organization of ProductionTask. The ontology defines three
types of tasks: (i) ComplexTask (either *disjunctive* or *conjunctive*); (ii) SimpleTask and;
(iii) Function. A ComplexTask is a ProductionTask (i.e., an instance of DUL:Method)
representing a compound logical operation. The hierarchical structure is enforced by
the property hasConstituent which associates ComplexTask with either SimpleTask or
other ComplexTask.

$$
\begin{aligned}
\texttt{ComplexTask} \sqsubseteq \quad & \texttt{ProductionTask} \sqcap \\
& \exists\,\texttt{DUL:hasConstituent.(ComplexTask} \sqcup \texttt{SimpleTask)} \sqcap \\
& \exists\,\texttt{DUL:isDescribedBy.OperativeConstraint}
\end{aligned}
\tag{2}
$$

A SimpleTask represents a leaf of the hierarchical structure of a ProductionMethod.
This concept describes primitive production operations that could be carried out leveraging
the functional capabilities of the agents. A SimpleTask requires the execution of a number
of Function instances by the agents.

$$
\begin{aligned}
\texttt{SimpleTask} \sqsubseteq \quad & \texttt{ProductionTask} \sqcap \\
& \exists\,\texttt{DUL:hasConstituent.Function} \sqcap \\
& \exists\,\texttt{DUL:hasConstituent.SimpleWorkpiece} \sqcap \\
& \exists\,\texttt{DUL:isDescribedBy.(InteractionModality} \sqcup \texttt{OperativeConstraint)}
\end{aligned}
\tag{3}
$$

The execution of a task should comply with operational constraints that are repre-
sented as ExecutionNorm. Two main types of execution norms can be defined: the concept
OperativeConstraint describes *norms* requiring the sequential or parallel execution of
tasks; the concept of InteractionModality instead characterizes *norms* about how agents
should cooperate to carry out a task.

### 3.3. Human Factor and User Model

The current work specifically focuses on the *Human Factor* context and elaborates
on its correlations with the *behavior* and the *production* contexts. Figure 4 shows part of
the taxonomic structure defined to represent behavioral and physical features of workers.
Such concepts define the variables composing the *user model* and therefore characterize
the representational space of qualitative aspects of a worker (i.e., types of DOLCE:Quality).

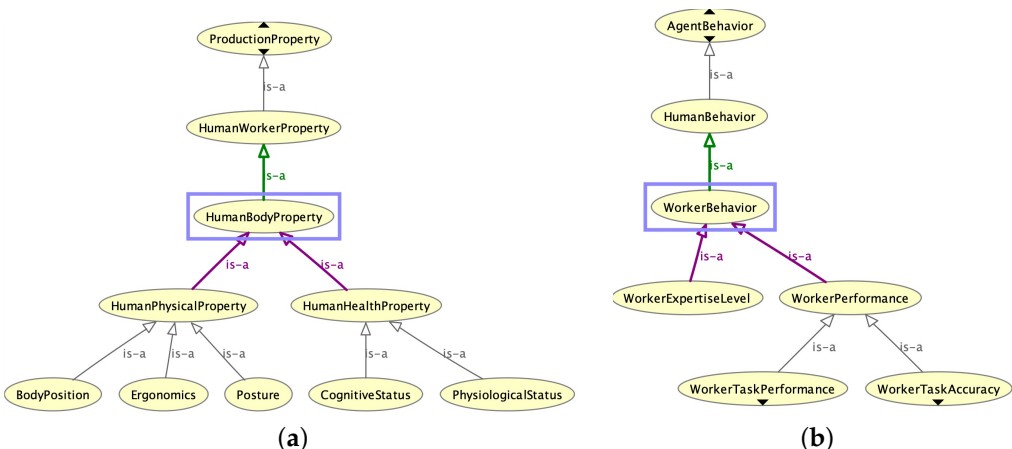

**Figure 4.** Excerpt of SOHO concerning the variables of the user model: (**a**) it shows concepts modeling qualities associated with the physical body of a worker; (**b**) it shows concepts modeling the qualities associated with the behavior of a worker.

Concepts characterizing the qualities of the physical body of a worker, Figure 4a, model physical, health, and cognitive parameters. Information about these variables enables the detection and monitoring of anomalous or dangerous working conditions, such as bad ergonomics, body position in hazardous areas or mental, and physical fatigue. Concepts concerning the qualities of the behavior of a worker, Figure 4b, instead model his/her performance in a given production scenario (e.g., the expertise level or the average time taken to perform a task).

The concept `WorkerExpertiseLevel` estimates "how much knowledgeable" a worker is about a particular production scenario. On the one hand, the expertise level determines the (sub)set of production tasks a human worker can carry out. For example, some tasks may require a certain minimum level of experience to be performed by a worker. On the other hand, it characterizes the reliability of the performance of a worker and thus the expected *uncertainty* about the duration of executed tasks. Low experience determines higher uncertainty and thus higher *variance* of the performance. High experience instead denotes lower uncertainty and thus more consolidated performance (i.e., lower *variance*).

The concepts `WorkerPerformance` supports a numerical representation of user performance. SOHO in particular distinguishes between *accuracy* (`WorkerTaskAccuracy`) and *efficiency* (`WorkerTaskPerformance`). These variables support the incremental definition of a *dataset* collecting historical data about performance. Such a dataset can be analyzed to incrementally *learn* performance of users and adapt collaborative processes over time. It can be used for example to infer an efficiency matrix encoding the average completion time of production tasks for (known) users.

## 4. User-Aware Collaboration

The production and user-centered knowledge is at the disposal of other modules to adapt production processes. Such knowledge is necessary to push forward novel collaboration paradigms where the system adapts interactions and collaborative processes to the *known* features of participating users. Knowledge inference and extraction procedures can be implemented to dynamically generate contextualized planning models to the specific features of a domain [19,20] as well as specific skills and preferences of a human worker. This section explains how the task planning, the Action and Motion, and the human–system interaction module take advantage of the user model to support personalization and adaptation.

Artificial Intelligence Planning and Scheduling [21–23] is well suited to endowing robot controllers with the *flexibility* needed to autonomously decide actions and adapt behaviors to the state of the environment [24,25]. Planning technologies generally pursue an *optimization perspective* aiming at finding plans that minimize or maximize a specific metric (e.g., minimization of the planning cost). Different metrics and features of a domain

can be taken into account, depending on the specific planning formalism used. In application domains such as HRC, reasoning about causality, time, concurrency, and simultaneous behaviors of domain features (e.g., the human and the robot) is crucial to synthesize and execute effective plans.

Task planning capabilities developed within SHAREWORK rely on the timeline-based formalism [26] and the PLATINUm software framework [27–29]. This planning formalism integrates reasoning about causal and temporal aspects of a planning problem and has been successfully applied to several concrete scenarios [30–32]. PLATINUm and the formalism introduced in [26] integrates *temporal uncertainty* and *controllability issues* to generate plans that are *robust* when executed in the real world [25,33]. Uncertainty is especially important in HRC where robots should continuously interact with *uncontrollable* autonomous entities such as human workers. Considering the manufacturing context and other works synthesizing optimal, multi-objective assembly processes [34,35], we extend PLATINUm by integrating multiple objectives and uncertainty. This allows us to synthesize (timeline-based) plans that achieve a good trade-off between efficiency (i.e., minimize the cycle time of collaborative processes) and safety (i.e., minimize the risk of collision), and take into account temporal uncertainty for reliable execution [36].

*4.1. Personalized Task Planning*

A timeline-based specification consists of several *state variables* that describe possible behaviors of domain features. A state variable is formally defined as a tuple $SV = \langle V, T, D, \gamma \rangle$. A set of values $v_i \in V$ represent states and actions the domain feature can assume or perform over time. A transition function $T : V \to 2^V$ specified valid sequences of values $v_i \in V$. A duration function $D : V \to \mathbb{R} \times \mathbb{R}$ specifies each value $v_i \in V$ the expected lower and upper bounds of its execution time. A controllability tagging function $\gamma : V \to \{c, pc, u\}$ specifies if the execution of a $v_i \in V$ is *controllable* (*c*), *partially controllable* (*pc*) or *uncontrollable* (*u*). Information about controllability allows a task planner to deal with uncontrollable dynamics of the environment when executing a (timeline-based) plan. This is known as the *controllability problem* [37] and is particularly important when an artificial agent such as a collaborative robot should interact with "unpredictable" agents such as a human worker. *Synchronization rules* constrain the "runtime" behavior of the modeled domain features. They specify causal and temporal constraints necessary to coordinate the different features as a whole complex system (e.g., a HRC cell) and synthesize valid temporal behaviors (i.e., the *timelines*).

The definition of state variables and synchronization rules modeling a HRC scenario follows a hierarchical decomposition methodology correlating high-level production goals to simpler production tasks and functions [38]. A state variable $SV_G$ describes the high-level production goals supported by the HRC work-cell. A number of state variables $SV_L^i$ where $i = 0, \ldots, K$ describe the production procedure at different levels of abstraction. The values of these state variables represent production tasks at a specific level of abstraction $i \leq K$ (where $K$ is the number of hierarchy levels of the procedure). A state variable $SV_R$ and a state variable $SV_H$ respectively describe the low-level operations (i.e., instances of `Function`) the robot and the human can actually perform. Finally, a set of synchronization rules $\mathcal{S}$ describes the procedural decomposition of high-level goals (i.e., values of state variable $SV_G$) into simpler production tasks (i.e., values of state variables $SV_L^i$), until they are associated with a number of functions of the human and the robot (i.e., values of state variables $SV_R$ and $SV_H$).

The state variable $SV_H$ describes behavioral dynamics of the worker collaborating with the robot. The state variable $SV_H = \langle V_H, T_H, D_H, \gamma_H \rangle$ is thus generated from the knowledge base according to the *user profile* of the participating worker. The values $v_j \in V_H$ are defined according to the tasks/functions the worker is actually able to perform in the given production scenario. No assumptions can be made on the actual duration of tasks/functions assigned to the worker. Consequently all the values of $SV_H$ are tagged as uncontrollable, $\gamma_H(v_j) = u, \forall v_j \in V_H$. The duration bounds of each value $v_j \in V_H$ and are defined

by taking into account the mentioned performance matrix that can be extracted from the knowledge base. A *performance vector* is extracted representing known performance of user $u_i \in \mathcal{U}$. Such a vector specifies, for each value $v_j \in V_H$, the average time $\delta_{i,j}$ the user $u_i$ takes to accomplish the task $task(v_j) = t_j \in \mathcal{T}$ ($\delta_{i,j} = \infty$ if no information is available).

At this point the *expertise level* of the user characterizes the expected variance of the average duration. The combination of this information is thus used to define the *personalized lower and upper duration bounds* for each value $v_j \in SV_H$. Specifically, a certain amount of uncertainty is associated to each of the three expertise levels defined into the ontological model: (i) *novice*; (ii) *intermediate*; (iii) *expert*. The higher the expertise level the lower the uncertainty about the performance. We define a *uncertainty index* associating each expertise level with constant value of uncertainty to consider: $\Omega = \{0.8, 0.5, 0.2\}$. Given a user $u_i \in \mathcal{U}$, a function $\Upsilon : \mathcal{U} \rightarrow \Omega$ specifies the uncertainty index corresponding to the expertise level of the user. The resulting duration bounds of the values composing the state variable of the worker $v_j \in V_H$ are then defined as follows:

$$D(v_j) = (\delta_{i,j} - \omega_i * \delta_{i,j}, \delta_{i,j} + \omega_i * \delta_{i,j}). \tag{4}$$

This mechanism dynamically adapt the temporal dynamics encapsulated into a task planning model according to the changing performance of the same worker as well as to the performance of different workers. The finer the temporal model of the worker, the better the optimization of plans and resulting collaborative processes [39].

### 4.2. Integrated Task and Motion Planning

To guarantee a high level of flexibility in the planning and execution of collaborative tasks, we deploy a hierarchical Task and Motion Planning framework that allows for online planning of the robot trajectories according to the user and the environment's state. This is key to ensuring a smooth collaboration between the human and the robot because the robot's tasks can be robust with respect to changes in objects and tools' positions, and the robot's movement can be optimized to avoid interference with the user's activities [40].

The key idea of our approach is that robot tasks coming from the task planner are symbolic and should be converted into a sequence of geometric movements by the Action and Motion Planning module. In this way, the task planner reasons about the best assignment and scheduling of tasks disregarding the actual geometric realization of each task. This is necessary because the robot's actual trajectories are not known a priori. Indeed, they need to be planned and adjusted on the fly according to the scene. Moreover, the motion planner can feature user-aware behavior that makes the robot's motion more dependable according to his/her preferences.

The Action and Motion planning module consists of a hierarchical framework composed of: (i) a task decomposition module; (ii) a proactive motion planner; (iii) a reactive speed modulation module. A scheme of the proposed framework is in Figure 5.

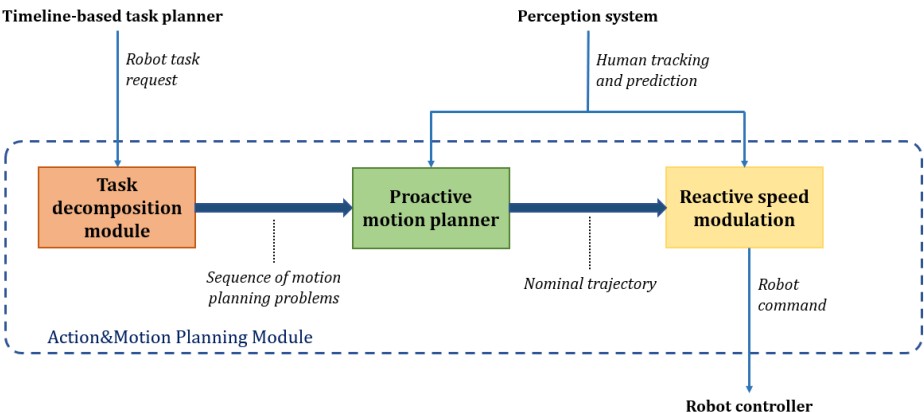

**Figure 5.** Action and Motion Planning module.

### 4.2.1. Task Decomposition Module

The task decomposition module owns a set of predefined skills; that is, high-level behaviors that the robot can execute autonomously (e.g., *pick an object*, *screw a bolt*). Skills are a model of an abstract task and allow the module to decompose a task into a sequence of robot movements. For example, the task *pick an object* is decomposed into a sequence of basic movements and conditions: (i) check if the gripper is empty; (ii) open gripper; (iii) move to approach pose; (iv) move to grasp pose; (v) close gripper.

Given an object to be picked, the task decomposition module retrieves the necessary geometric information from the scene (e.g., by querying the Knowledge Base), checks whether the task conditions hold, and initializes the basic movements according to the scene state. Notice that, at this stage, a task might have multiple equivalent geometric realizations. For example, the symbolic task *pick a blue cube* may require choosing among multiple blue cubes, each with numerous grasping points. This level of complexity is addressed by the *proactive motion planner*.

### 4.2.2. Proactive Motion Planner

The proactive motion planner solves the motion planning problem related to each basic movement of a task, as decomposed by the Task decomposition module. The term proactive distinguishes this module from the reactive speed modulation module. The proactive planner is intended to find a collision-free trajectory according to a prediction of the user's actions and movements. Once a trajectory has been found, its execution starts, and the reactive layer monitors and adjusts it according to real-time scene information. Moreover, the path is sent to the Human-System Interaction Module for visualization so that the user will foresee the robot's movement in the short run.

The proactive trajectory planner has been implemented by using the standard path-velocity decomposition paradigm, in which a path planner finds a collision-free path from a start to a goal state, and a path parametrization algorithm (e.g., TOPP [41]) optimizes the velocity profile along the path. Regarding path planners, sampling-based algorithms are preferred, for they can efficiently deal with high-dimensional search space [42].

User awareness is embedded in the path planner using a cost function that depends on the human state. The typical approach minimizes a weighted sum of an efficiency term (e.g., the path length) and user-aware terms, such as human–robot distance [43], trajectory repeatability [44], or human visibility [45].

### 4.2.3. Reactive Speed Modulation Module

The reactive speed modulation module modifies the nominal speed during the execution of each trajectory. This is necessary to meet safety requirements and avoid unnecessary safety stops. In general, reactive motion planners are shifting from simple yet conservative strategies such as safety zones to optimized methods that adapt the robot motion continuously [46,47].

In this work, we adjust the robot's speed according to the safety requirements imposed by safety specifications. In particular, the technical specification ISO/TS 15066 (*Robots and robotic devices—Collaborative robots*) [48] defines speed reduction rules for collaborative operations with and without admissible contact between robots and humans. For example, if *speed and separation monitoring* is applied, the human–robot distance $S$ must not fall below a protective distance $S_p$.

To satisfy the safety requirement without jeopardizing the smoothness of the collaboration, we adopt a continuous modulation of the robot speed. The robot nominal velocity is scaled at high rate by a speed override factor $s_{\mathrm{ovr}} \in [0,1]$ according to the following rule:

$$s_{\mathrm{ovr}} = \min \left( \frac{v_{\max}}{v^{rh}} , 1 \right), \tag{5}$$

where $v^{rh}$ is the human–robot relative speed and $v_{\max}$ is the maximum allowed relative velocity, derived from (6) by using the measured user position and velocity in the cell, that is:

$$v_{\max} = \sqrt{v_h^2 + (a_s T_r)^2 - 2a_s\big(C - S(t_0)\big)} - a_s T_r - v_h, \tag{6}$$

where $S(t_0)$ is the human–robot relative distance at the current time instant, $v^r$ is the robot velocity toward the human, $v_h$ is the human velocity toward the robot, $a_s$ is the maximum Cartesian deceleration of the robot toward the human, $T_r$ is the reaction time of the robot, and $C$ is a parameter accounting for the uncertainty of the perception system.

### 4.3. Augmented Human–Robot Interaction

The human–robot interaction framework aims to structure a usable and personalized interaction pipeline between the operator and the robot towards increased awareness and well-being. All those attributes are mostly accomplished by using multiple human senses (i.e., vision, hearing, touch) through interaction modalities available in customized interfaces [49]. Both modalities and customization options formulate a user-centric framework that can meet the requirements of novice and advanced operators.

In terms of architecture, the multi-modal interaction framework consists of three layers (see Figure 6). A broker forms the top layer of this module and is responsible for parsing information from the SHAREWORK's modules to the end devices, and vice versa [50]. The intermediate layer incorporates the available end devices, thus their respective applications. The bottom layer gathers all the supported interaction modalities based on the specifications of the intermediate hardware. The existence of the broker ensures stability against a varying number of deployed devices. During operations, there are redundant ways of interaction since information can flow simultaneously to all devices. This suppleness does not only serve the anthropocentric SHAREWORK system design principles, but also contributes to the overall system resilience against hardware limitations (e.g., battery, network range) or even issues (e.g., damages).

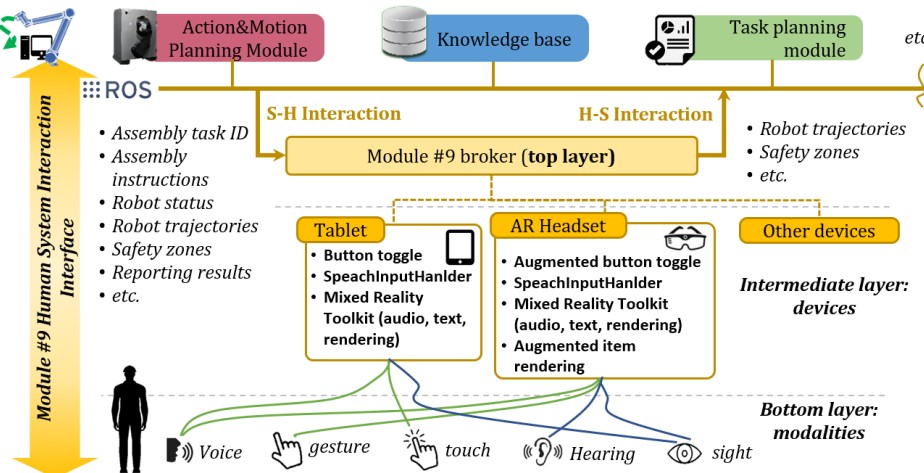

**Figure 6.** SHAREWORK's Human-System interaction module architecture.

Focusing on the information streams, the module comprises mechanisms for human–system (HS) and system–human (SH) interaction. The former ones are needed either for operator monitoring or direct robot control. Despite the advances in machine learning for human activity recognition, the improvisation of operators can still highlight limitations in those systems. Thus the developed interaction framework supports functionalities for monitoring purposes. In detail, all deployed applications involve "Task completed" feedback apparatus in the form of voice commands, touch, and augmented buttons. The same modalities can also be used as inputs for direct robot and system control (e.g., stop or proceed). Each application processes those inputs and communicates to the broker normalized commands or requests that are parsed to the rest

of the SHAREWORK modules through ROS. On the contrary, when the system communicates to the human, the involved modules share information to the broker that is then streamed simultaneously to all end devices. Each application makes available the information based on the hardware's capabilities in the form of textual, graphical, or audio material.

The volume and type of communicated information are closely related to the operator's experience level. For novice users, the module offers intuitive visual instructions that can support them during assembly operations via augmented 3D models, panels, arrows, or screen-based figures. For greater awareness, robot-centric information can also be provided through 3D augmented trajectories, notifications, and warnings. Textual instructions and info are standard in plain or extended format. Unlike novice users, who need support and a clear description of robot behavior, experienced operators could be distracted if they are communicated with all aspects of information. For this reason, the customization of the interaction framework can be performed during runtime through the related options panels. According to each operator's entity, tailoring of the interfaces is supported by the Knowledge Base. The customization options suggest the selection of available devices, available modalities, assembly information detailing, feature positioning, button positioning, and robot information detailing. The personalization of the system's front-end through customizable applications and the selection of multiple devices is achieved by implementing a distinct hierarchical architecture.

### 5. Integration and Deployment: A Case Study

We demonstrate the proposed approach in a case study derived from a mechanical machining scenario. The case study is characterized by unpredictable market changes in terms of demand, which require massive use of Flexible Manufacturing Systems (FMSs) to remain highly competitive in the market [51]. In FMSs, parts to be machined are mounted on multi-fixturing devices called pallets. Pallets are manually assembled at a loading/unloading station (LUS) and moved from/to general-purpose machine centers to be machined. The number of pallet configurations, i.e., pallet mounting clamping systems/jigs, and products present simultaneously in an FMS can be considerable. Due to the high number of different operations to be performed on the pallets, LUSs influence FMS performance in terms of final throughput. Specifically, three critical operations at LUS are performed: assembly, disassembly, and quality inspection. The application is stimulating for human–robot collaboration because the process throughput would benefit from juxtaposing humans' manipulation skills and robots' tirelessness. For example, robots could be exploited to perform batches of simple, repetitive operations, while a human could perform the most complex operations and perform quality checks. Note that no fixed scheduling usually applies [52]; a dynamic online reconfiguration of the workflow is required by operators who may change sequences and roles.

Within this context, we consider the scenario shown in Figure 7. A collaborative LUS is composed of a small-size robot, Universal Robots UR10e, mounted on a linear track to extend its range of motion. The LUS owns four pallet positions: P0 is the arrival position of a new pallet brought by a mobile robot; P1 and P2 are the working position, where the pallets are mounted, unmounted, and checked; P3 is the departure position, where a mobile robot will load the finished pallet to move it to next stage of the process. The robot and the human operator can work simultaneously at the LUS, either on the same pallet or two different pallets. The process requires the following stages:

1. A mobile robot brings a new pallet to P0;
2. The pallet is moved to a free position (P1 or P2). Notice that the pallet can be moved to P2 only if P1 is not occupied;
3. The pallet is unmounted to extract the finished part;
4. A new raw part is inserted, and the pallet is mounted;
5. The pallet is moved to P3;
6. A mobile robot picks up the pallet from P3.

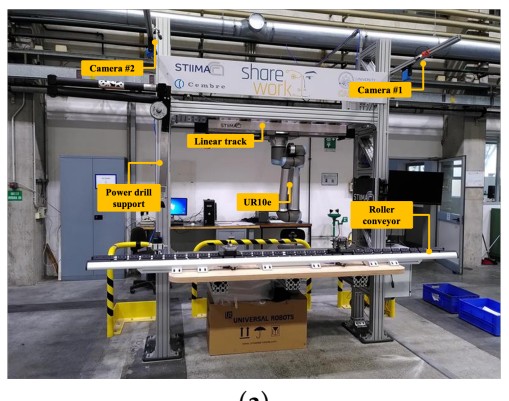 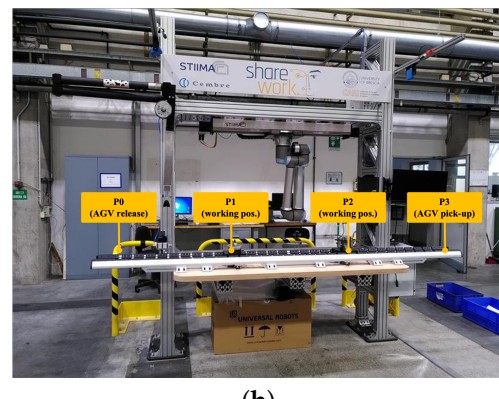

(**a**)          (**b**)

**Figure 7.** Demonstrator setup and description of the work positions: (**a**) Setup components; (**b**) Setup working positions.

Steps 2 and 5 are always performed by the human operator because the robot is not able to lock/unlock and move the pallet. Steps 3, 4, and 5 can be performed by both the robot and the human. Notice that more than up to four pallets can be present at the LUS at the same time, meaning that Steps from 2 to 5 can be performed without a fixed scheduling and assignment wither by the human or the robot. Moreover, pallets can have different geometries and therefore requires different operations to be mounted and unmounted. In this case study, we consider three different types of pallets requiring high flexibility in the planning and execution phases.

### 5.1. Process Representation in the Knowledge Base

To successfully coordinate human and robot operations, it is necessary to configure the *Knowledge Base* module of Figure 2 first. This configuration step allows the system to build an abstraction of production procedures characterizing the specific needs/requirements of an HRC cell and the specific skills and features of participating acting agents. To this aim, we manually define an ontological model of the scenario using Protégé (A well-known editor for ontologies and knowledge bases—https://protege.stanford.edu (accessed on 21 June 2022)). We define individuals and assert properties necessary to characterize the (relevant) information about the *production environment* and the *capabilities* of the agents that take part in the process.

The main elements of the environment are the workpieces (i.e., *pallets*), the worker and the cobot agents, and the positions they occupy while performing operations. Workpieces can be of three types entailing different geometric constraints and low-level operations for their manipulation. These workpieces are thus modeled separately as three distinct instances of `Workpiece`: (i) 0218; (ii) 1121; and (iii) 1122. This distinction supports the contextualization of production procedures according to the particular type of workpiece to be worked. During the execution of a production procedure, each workpiece occupies a specific environment location. In the considered scenario, such environmental locations are subject to *physical constrains* limiting the number of objects that can occupy them simultaneously. They are thus modeled as `BinaryProductionLocation` that are `ProductionLocation` associated with a `ResourceCapacity`, which limits to 1 the number of `ProductionObjects` that can stay at the same location simultaneously (i.e., these locations are characterized by a *binary* state denoting the location as *free* or *busy*).

Each type of workpiece is associated with a `ProductionGoal` specifying a different `ProductionMethod` and different production operations. Such production operations are defined as individuals of `ProductionTask`. The knowledge base describes *operational constraints* and *alternative decomposition* of tasks as well as *alternative assignments* to the human and to the robot. In this regard, individuals of `DisjunctiveTask` describe alternative way of implementing/decomposing a `ProductionTask`. For example, the general task `process_1121` is modeled as `DisjunctiveTask` and is associated with two alter-

native sub-tasks through the property `DUL:hasConstituent`: (i) `process_1121_p1` and; (ii) `process_1121_p2`. Both sub-tasks are instances of `ConjunctiveTask` and represent two alternative ways of performing the production task `process_1121_p2`: (i) perform production operations for workpiece 1121 on `position1`; and (ii) perform production operations for workpiece 1121 on `position2`. The actual choices would be made dynamically by a task planner depending on previously scheduled operations and the *known* state of physical locations/positions of the HRC cell.

A similar decomposition is defined for low-level tasks that can be assigned to the human or to the robot. An example is the operation requiring to mount the pallet 1121 in a specific position of the cell. The `DisjunctiveTask mount_1121_p2` is decomposed into two (alternative) simpler `ProductionTask` that are: (i) `mount_1121_p2_worker` and; (ii) `mount_1121_p2_cobot`. This disjunction characterizes the alternative choice of assigned the mounting task to the worker or to the robot. The two sub-tasks are both instances of `IndependentTask` meaning two individuals of `SimpleTask` associated with a `CollaborationModality` of type `Independent`. Following the ontological definition of *independent collaborative tasks*, they are respectively decomposed into a `HumanFunction` and a `RobotFunction` of type `Assembly` representing the actual operations performed on the workpiece.

The defined knowledge base completely characterizes the production process and can be used to configure the planning and interaction modules deployed into the scenario. A designed *knowledge extraction procedure* automatically generates *contextualized* timeline-based specifications for: (i) hierarchical decomposition and planning constraints concerning known goals; and (ii) temporal dynamics and controllability properties associated with robot and worker capabilities. Such specification provides the Task Planner with the rules to compute collaborative plans for the considered manufacturing scenario at hand. A graph-based description of production procedures is automatically extracted from the knowledge base and used to generate a suitable timeline-based task planning model [19,20]. The resulting production procedure is organized into several *hierarchical levels* correlating high-level *production goals* with low-level tasks and individuals of `Function` the human and the robot should perform to carry out related production processes correctly. The following section describes with further detail the timeline-based model and provides an example of a plan.

### 5.2. Task Planning and Scheduling

A timeline-based task planning model is synthesized by the knowledge base to "operationalize" production procedures and coordinate human and robot behaviors. A number of *state variables* are defined to characterize states and/or actions that relevant domain features assume and/or perform over time. Four state variables $SV_{p0}$, $SV_{p1}$, $SV_{p2}$, $SV_{p3}$ describe the state of the working positions of the pallets. Since these physical locations are modeled as `BinaryProductionLocation` in the knowledge base, these variables are associated with two values, $V_{p0}$, $V_{p1}$, $V_{p2}$, $V_{p3} = \{Free, Busy\}$. Then, we define transitions $T_{p0}(Free)$, $T_{p1}(Free)$, $T_{p2}(Free)$, $T_{p3}(Free) = Busy$, $T_{p0}(Busy)$, $T_{p1}(Busy)$, $T_{p2}(Busy)$, $T_{p3}(Busy) = Free$ and duration $(1, +\infty)$ for all of them. These state variables are used to encode *binary resource constraints* and, thus, enforce a mutually exclusive use of the associated physical locations.

Other two state variables describe possible behaviors of the human and the robot in terms of the set of `Function` that they can perform over time. These functions are the *inferred* instances of low-level operations the human and the robot can perform in the considered scenario. The state variable of the robot $SV_R$ is thus a associated with the set of values denoting the function it is supposed to perform $V_R = \{Idle, Release\_p1, Release\_p2, Pick\_p1, Pick\_p2, Assemly\_p1, Assembly\_p2, Disassembly\_p1, Disassembly\_p2\}$. The values $Release\_p1$, $Release\_p2$ and $Pick\_p1$, $Pick\_p2$ are instances of the function `PickPlace` and denote respectively the operations of removing a worked piece from the pallet (i.e., *release piece*) and placing a new raw piece into the pallet (i.e., *pick piece*). The transition function requires all value changes to pass through the idle state as follows: $T_R(Idle) \in \{Release\_p1, \ldots, Disassembly\_p2\}$; $T_R(Release\_p1) = \{Idle\}$; $\ldots$; $T_R(Disassembly\_p2) =$

$\{Idle\}$. All the values of the robot $v_{R,i} \in V_R$ are tagged as *partially controllable* $(\gamma_R(v_{R,i}) = pc)$ because the actual duration of their execution can interfere with the worker. The duration bounds of these values instead is set according to the average observed execution time. The state variable of the human $SV_H$ is structured similarly to $SV_R$. In this case, it is necessary to consider the additional operations the worker can perform and robot can not. These are modeled with additional values $V_H = \{PickPlace\_p0p1, PickPlace\_p1p2, PickPlace\_p2p3, \dots\}$. The value transition function follows the same "pattern" of $SV_R$. However, in this case, all the values $v_{H,i} \in V_H$ of the state variable $SV_H$ are tagged as *uncontrollable* $(\gamma_H(v_{H,i}) = u)$ since the system cannot control the behavior of the worker. Furthermore the duration bounds of the values are defined according to Equation (4) and thus they depends on both the average duration of their execution and on the *uncertainty index δ* set according to the *expertise level* of the worker.

To synthesize production operations, it is necessary to define "functional" state variables encapsulating abstract production tasks. Such state variables are directly associated with the production procedure extracted from the knowledge base. The actual number of these variables (and their values) depend on the complexity of the modeled procedure. In general, each "production" state variable is associated with a specific *abstraction level* of the extracted hierarchical procedure. A *goal state variable* $SV_G$ encapsulates high-level production requests and is associated with the individuals of `ProductionGoal`. Individuals of this concept are generally root elements of the production procedure and are mapped to the values of $SV_G$. In this case we have three different types of goals, each associated to a particular type of pallet $V_G = \{process\_1121, process\_1122, process\_0218\}$. Three different hierarchical procedures correspond to these three goals. These values are all *controllable* $(\gamma(v_{G,i}) = c)$ and do not have specific duration bounds since their actual duration depends on the planning and scheduling of underlying human and robot operations. Intermediate $N - 1$ levels of the procedure are modeled through "production state variables" $SV_{L_1}, \dots,$ $SV_{L_{N-1}}$. The last hierarchical level $(N)$ of the decomposition entails individuals of `Function` that are already represented through $SV_R$ and $SV_H$. The values of production state variables represent individuals of `ProductionTask` such as *unmount\_1121\_p2*, *mount\_1121\_p2* and thus complex/abstract production operations that need to be further decomposed in simpler ones. Starting with high-level production requests (i.e., values of the goal state variable $v_j \in V_G$) task decomposition and the needed causal and temporal constraints are modeled through a set of *synchronization rules*. Each rule has individuals of `ProductionTask` (i.e., values $v_{L_i} \in V_{L_i}$) as trigger (i.e., the *head* of the rule). Individuals of `DisjunctiveTask` are triggers of different rules in order to model alternative decomposition.

The Task Planning Module implements *goal-oriented acting* capabilities using the opensource ROSJava Package ROXANNE (https://github.com/pstlab/roxanne_rosjava.git (accessed on 21 June 2022)). Once configured, the module is ready to receive production requests (i.e., *planning goal*) through a dedicated input topic. The synthesis of a task plan consists in deciding the assignment of production tasks to the human and the robot that best takes advantage of the collaboration (i.e., optimize the production process) in the given scenario [7,28,39]. The resulting assignment is then dispatched online to the human and to the robot by sending task execution requests respectively through the Human-System Interaction Module and the Motion Planning Module (see Figure 2).

Figure 8 shows an example of a timeline-based plan. It specifically shows the timelines of the plan through a Gantt representation depicting *tokens* planned for the state variables of the domain and their allocation over time. Note that this Gantt representation shows a specific instance of the plan called the *earliest start time*. Timelines indeed encapsulate an envelope of possible temporal behaviors (i.e., possible plan instances) through *temporal flexibility* [26]. This flexibility is crucial to deal with *temporal uncertainty* and support *reliable execution of timelines* in real environments [39,53].

| | 0 | 1 | 2 | 3 | 4 | 5 | 6 | 7 | 8 |
|---|---|---|---|---|---|---|---|---|---|
| Goal | Goal(1122) | | | | | | | | |
| Process | Disassembly(1122, p1) | | | | Assembly(1122,p1) | | | | |
| Tasks | Unmount(p1, r) | | Release(p1, r) | | PickPlace(p1,r) | | Mount(p1,h) | Finish(p1,h) | |
| Worker | Idle() | | | | | | Mount(p1) | Idle() | PickPlace(p1,p3) |
| Cobot | Unmount(p1) | | Idle() | PickPlace(p1,rbox) | Idle() | PickPlace(pbox,p1) | Idle() | | |

**Figure 8.** Simplified view of a plan synthesized for the execution of a collaborative process concerning `Workpiece` 1122. It shows the timelines synthesized for each state variable of the task planning model (i.e., *Goal*, *Process*, *Tasks*, *Worker* and *Cobot* state variables) with the scheduling of related tokens.

### 5.3. Action Planning and Execution

High-level tasks dispatched by the task planner module are put in place by the action planning module. This module converts symbolic tasks into a sequence of robot movements and tool operations (e.g., open/close gripper). The *Task Decomposition* module receives the task from the task planner and queries a database to decode the type of the task and its geometrical properties. The type of task and its properties determine the set of operations that a task requires.

When a task request comes from the task planner, the Task Decomposition module converts it into a set of basic operations. For example, task *mount_1121_p2* boils down to: (a) move to $P_2$ approach position; (b) approach nut; (c) unscrew nut (activate power drill); (d) push locking bracket; (e) move to piece grasping pose; (f) close gripper; (g) move to unloading box; (h) open gripper. Each operation corresponds to a point-to-point robot movement or a change in the state of the robot's auxiliaries (e.g., the gripper and the power drill). For all robot's movements, the *Task Decomposition* sends the decomposed actions to a motion planning algorithm. When the task is executed, it returns the outcome to the task planner. If the task is successful, the task planner will dispatch the next task in the plan. Otherwise, it would replan according to the reported error. Notice that, since the proposed framework is designed for dynamic environments, task decomposition and motion planning are performed online, based on the current state of the cell.

The *Task Decomposition* module was developed in C++ and Python 3 within the ROS framework. The communication with the task planner is managed by a ROS-action server that receives the tasks from the task planner and queries a MongoDB database to retrieve the task properties. The planning and execution phases are managed by *manipulation* framework, an open source library that implements basic skills [54]. The *manipulation* framework uses the *MoveIt!* planning pipeline and planning scene. Thanks to *MoveIt!*'s plugin-based architecture, it is possible to load motion planners dynamically from state-of-the-art libraries available in *MoveIt!* (e.g., OMPL, CHOMP, and STOMP). In this work, we use a human-aware path planner [55], which accounts for the position of the operator in the cell, according to Section 4.2.2. The *manipulation* framework is also modular with respect to the controller. In this work, we implemented the human-aware reactive speed modulation module (see Section 4.2.3) as a ROS controller that changes the robot speed according to the human–robot relative distance. This allows for a real-time implementation with a sampling rate equal to that of the robot controller (500 Hz), ensuring prompt reaction of the robot motion.

### 5.4. Human-System and System-Human Interaction

For this industrial case, the HS-SH interaction module was deployed by spawning two applications, hosted on Augmented Reality Headset Microsoft HoloLens 2 and Android Tablet Samsung Galaxy S4 (Figure 9). Voice, gesture, touch, hearing, and sight-related modalities are available during operation, either for direct system control or for worker support. The Knowledge Base configures the type of modals and features within the application environments according to the operator's level of expertise. Online customization options are offered to users to maximize personalization thanks to several options for each feature. Authentication via operator profiles ensures that user models are updated with the customization settings and are linked to each operator.

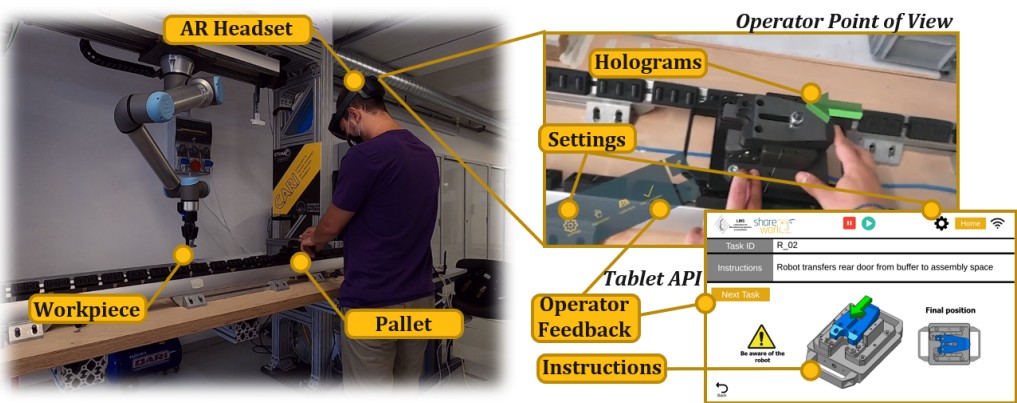

**Figure 9.** Demonstrator setup and HS-SH interaction module.

The human worker and the robotic arm are aware of each other through bilateral communication of information about each agent's actions. More specifically, the user can press easy-to-use buttons to send feedback to the Task planner about the successful execution of a human task or action. On the contrary, the robot's status is broadcasted via textual panels in addition to visualized robot trajectories in 3D augmented-reality (i.e., AR headset) or 2D screen-based (i.e., tablet) formats, as planned by the Motion Planner. Awareness about robot actions is also promoted via audio notifications that are enabled upon robot movements.

The implemented interaction module also supports users during manufacturing operations through intuitive instructions in extensive or plain form, depending on their preferences and expertise. The AR application augments the physical system by visualizing digital assistive content within the workstation (Figure 10). In detail, 3D augmented models and arrows (static or moving) instruct the operator on how to manipulate related components toward successful assembly. On the same basis, the tablet application provides assistive figures. In parallel, task information panels are filled by the "Task planner", providing Task id, name, remaining tasks, and instructions about current operation in both applications.

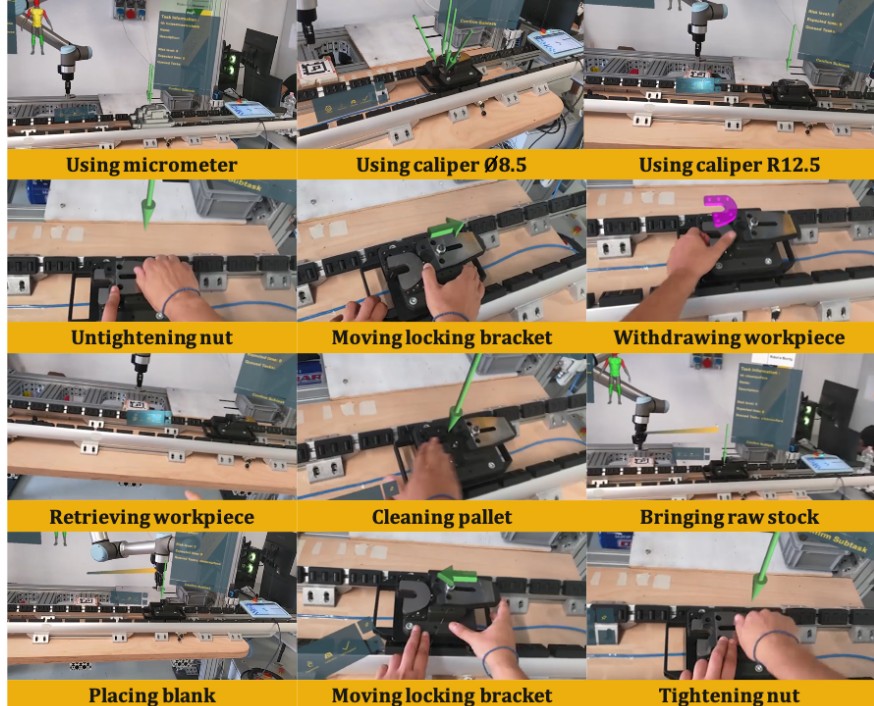

**Figure 10.** Operator point of view for indicative tasks (AR application).

*5.5. Results and Discussion*

In general, cyber–physical systems entail the need for intuitive human–system interfaces. This is even more crucial in anthropo-centric human–robot collaboration systems that incorporate online task and motion planning mechanisms. Indeed, these interfaces help promote mutual awareness about collaborative agents' current and future actions. Focusing on the human operator(s), the exposure to robots and safety-related information (e.g., future robot tasks, future trajectories), as planned from the related modules, can contribute to an increased sense of safety, feeling more trust in the system, and improved overall confidence during the fence-less coexistence. Similarly, highly usable monitoring and interaction mechanisms can support the planning and system adaptation modules on accurate and correct decision making due to reliable scene reconstruction of the cell's status.

In this sense, the proposed framework brings together different solutions coming from the state of the art in knowledge reasoning, planning, and communication to advance the readiness of HRC research solutions in real-world industrial problems. In particular, the integration of reasoning and profiling capabilities connected to robot planning capabilities is crucial to enhance robotic solutions' flexibility and efficacy in real-world manufacturing. Additionally, complete and well-structured knowledge about production scenarios can be used to smoothly adapt collaborative plans and related robot motions to different skills and production operational and safety requirements. This increased level of *awareness* and adaptation of robots is achieved dynamically by combining the developed AI-based cognitive capabilities into novel controllers. Namely, integrating such cognitive capabilities is crucial to support higher level of flexibility and allow robots to autonomously contextualize their behavior to production needs and personalize collaboration according to workers' skills and preferences.

It is worth stressing that the proposed framework is intended as a flexible toolkit of modules to be used based on the application requirements. For example, in a preliminary version of the framework [10], we addressed a use case inspired by the automotive industry. In that case, we did not use online Action and Motion planning because the application required a heavy-duty robot to perform slow and repetitive trajectories. This confirms the generality of the approach. At the same time, modularity represents a key advantage in the hand of the system integrator.

This work focused on the integration of different technologies at a single-cell level. An open point (to be considered in future works) is how to scale the proposed framework up to the shop floor level. Collaborative cells and operators should be aware of a relevant operation running in other cells or storage status. This requires additional effort from the planning and the communication points of view. In terms of planning, the Knowledge Base of each cell should communicate with a supervisory software system of the plant and leverage this augmented knowledge to plan shop floor-aware actions. The system should also inform the operators of relevant features they could exploit in the middle- or long-term reasoning. In this regard, research trends concerning developing a *digital twin* of factories [56–58] seem quite correlated with the current work. The proposed AI-based representation and control capabilities would strongly benefit from integrating a rich and updated digital representation of the whole shop floor/factory. The digital twin would indeed represent a precious source of valuable knowledge to (locally) enhance the awareness of robot controllers about the current (and future) production and enable more contextualized planning/control decisions. A further open point concerns how to combine the Planning and Execution flexibility desired in most applications with industrial safety requirements. As shown in our framework, flexibility means that the system may change the way the system acts (e.g., modifying the sequence and the time of tasks to be executed). Risk assessment procedures currently used in the industry must catch up with this re-configuration flexibility. Online and dynamic risk assessment appears a fundamental aspect to be addressed to increase the technology readiness level in the next future (see, e.g., [59]).

## 6. Conclusions and Future Works

This work shows the research activity conducted within the EU-funded project SHARE-WORK to foster *user-awareness* in HRC scenarios. The work proposes a methodology to deploy control architecture based on the integration of recent advancements in knowledge representation , task and motion planning, and human–system communication. We have shown that the integration of these cutting-edge technologies lays the groundwork to push a change of paradigm in human–robot collaboration towards contextualized and user-centered production processes. The developed technological modules have been successfully deployed on a real-world manufacturing scenario showing the technical feasibility of the approach. Future works will investigate the applicability of the proposed approach at a shop floor level and with users with different skills and features.

**Author Contributions:** Conceptualization, D.A., M.F. and A.U.; methodology, D.A., M.F. and A.U.; software, D.A., M.B., M.F., N.F., P.S.K., S.K. and A.U.; validation, D.A., M.B., M.F., N.F., P.S.K., S.K. and A.U.; formal analysis, M.F. and A.U.; investigation, M.F. and A.U.; resources, A.S. and P.T.; writing—original draft preparation, D.A., M.F. and A.U.; writing—review and editing, M.F. and A.U.; supervision, S.M., N.P., A.O. and A.Z.; project administration, S.M., N.P., A.O., A.C. and A.Z.; funding acquisition, S.M., N.P., A.O., A.C. and A.Z. All authors have read and agreed to the published version of the manuscript.

**Funding:** This work was partially funded by EU project SHAREWORK - H2020 Factories of the Future GA No. 820807.

**Conflicts of Interest:** The authors declare no conflict of interest. The funders had no role in the design of the study; in the collection, analyses, or interpretation of data; in the writing of the manuscript, or in the decision to publish the results.

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
