# Peer review of "Design of Advanced Human–Robot Collaborative Cells for Personalized Human–Robot Collaborations"

_applsci, doi:10.3390/app12146839_

Round 1
Reviewer 1 Report
This is a very interesting case study of the EU Project in the field of work sharing between man and machine while maintaining safe working conditions.
The background of the study is outlined, research objectives are specified, and described hypotheses that have been tested. Methodology of the study is presented and explained. Results section is a concise summary of the most important results that have been obtained. The main conclusions are indicated. Authors explain why the study results are significant and provide the key take-home message.
The only remark concerns fig. 8, 9 which are illegible. If they will be enlarged via technical edition, the paper can be published in the present form.
Reviewer 2 Report
a) The paper is well written and requires only few check spell corrections:
1.- Line 195: “the the”
2.- Line 369 “ a start to a goal strat”
3.- Line 476: “wither” should be “either”
4.- Line 579: Equation 4 must be between parenthesis
b) The paper is clear and presents a methodology for design of a collaborative human-robot cell, at the beginning the papers seems like a survey and it is not very clear the contribution of the paper, that seems to be the integration of several methodologies, this is a somehow presented at the results section, however advantages of the proposed methodology are not clear , and it is hard to compare to other proposals.
c) Figure 8 is no readable.
Reviewer 3 Report
ID: applsci-1782965-peer-review-v1
Title: Design of Advanced Human-Robot Collaborative Cells for Personalized Human-Robot Collaborations
The focus of the paper is on a Human-Robot Collaboration cell which requires a continuous peer-to-peer interaction between humans and robots. For the sake of justification, authors leveraged a manufacturing case study of the robotic cell to show a possible deployment of the proposed framework in a real-world industrial scenario.
The topic can be interest of robotic and manufacturing researchers. If you ask me, it has a good chance of publication as the paper is well written and clearly have use cases in industry. However, I have some comments about content of the paper, especially its contribution. Please see the following comments:
- My main concern is about Figure 1, which shows a high-level overview of the SHAREWORK architecture. It is depicting the set of modules and the high-level flow of information among them. Modules of the architecture include the Knowledge Base Module, the Task Planning Module, the Action&Motion Planning Module, and the Human-System Interaction Module. However, I cannot get the main contribution of the architecture. Since it is an important figure in this paper, authors should clearly explain the contribution of this architecture.
- The first time authors use an abbreviation in the text, they should present both the spelled-out version and the short form. When the spelled-out version first appears in the narrative of the sentence, put the abbreviation in parentheses after it. Example: Page 2: In this regard, AR is an imposing …
- Never use etc. at the end of a series that begins with for example, e.g., such as, and the like, because these terms make etc. redundant: they already imply that the writer could offer other examples.
Page 4: e.g,. robot capabilities, operator skills and performance profile, production goal decomposition, etc.
Page 11: e.g., stop, proceed, etc.
Page 18: i.e., future robot tasks, future trajectories, etc.
- Figure 6 have never addressed in the body of the paper. Please check all figures and tables on this matter to avoid any inconsistency in the body of the paper.
- The resolution of some figures is low and they are unclear. Specifically, this is about Figures 3, 8 and 9. They are important figures, and so please make their resolution higher. As Figure 8 is a large one, I think authors should find a better way to represent it.
- Regarding industrial applicability of the FMS in Section 5 for mass production, I also think a brief literature review is essential to justify the use case of the cell in mass production. It is about assembly (production) and quality inspection in HRC cells. So, some papers on (robotized) production line without (with) inspection can be cited as a potential application of methods: [a] Pure cycles in two-machine dual-gripper robotic cells, RCIM, vol. 48, pp. 121–131 [b] resolution of deadlocks in a robotic cell scheduling problem with post-process inspection system: avoidance and recovery scenarios, IEEE International Conference on Industrial Engineering and Engineering Management (IEEM), 2015, pp. 1107-1111
- The title of some papers in the reference list is lowercase, whereas the others are Capitalized each Word. This is also a case of inconsistency. Please make all of them lowercase for the sake of simplicity.
- The paper will gain maximum benefit from a proofread.
Page 2: Driven by these consideration --> Driven by this consideration
Page 3: based on the the data received --> based on the data received
Page 3: converts an high-level task --> converts a high-level task
Page 4: e.g,. robot --> e.g., robot
Page 6: three types of task --> three types of tasks
Page 8: Consequently the user --> Consequently, the user
Page 14: Similar disjunctive decomposition are defined --> Similar disjunctive decompositions are defined
Page 14: Function the can perform over time --> Function that can perform over time
Page 15: To synthesize production operations it is --> To synthesize production operations, it is
Page 19: We shown that --> We have shown that
Reviewer 4 Report
A list of assumptions followed in the general architecture should be enlisted.
Inter connections between sub-tasks should be given (for example scheduling, task planning, motion planning how there are connected, and how the data flows?)
For a specific assembly process, would it (dynamic task and motion planning( be cost-effective?
Please specify the AR/other devices in figure 1.
Please see some recent literature on human-robot collaboration and robotic task planning(for section 5.2 of the manuscript) .
1. Inkulu, Anil Kumar, et al. "Challenges and opportunities in human robot collaboration context of Industry 4.0-a state of the art review." Industrial Robot: An International Journal 49.2 (2022): 226-239.
2. Sheridan, Thomas B. "Human–robot interaction: status and challenges." Human factors 58.4 (2016): 525-532.
3. Murali, G. Bala, et al. "Optimal robotic assembly sequence planning using stability graph through stable assembly subset identification." Proceedings of the Institution of Mechanical Engineers, Part C: Journal of Mechanical Engineering Science 233.15 (2019): 5410-5430.
4. Ren, Weibo, et al. "The decision-making framework for assembly tasks planning in human–robot collaborated manufacturing system." International Journal of Computer Integrated Manufacturing (2022): 1-19.
5. Bahubalendruni, M. R., & Biswal, B. B. (2017). A novel concatenation method for generating optimal robotic assembly sequences. Proceedings of the Institution of Mechanical Engineers, Part C: Journal of Mechanical Engineering Science, 231(10), 1966-1977.
6. Malik, Ali Ahmad, and Arne Bilberg. "Complexity-based task allocation in human-robot collaborative assembly." Industrial Robot: the international journal of robotics research and application (2019).
Please improve the quality of Figures 3,8.
The completeness of the proposed methodology for its applicability to different products(assembly) or processes would further enhance the quality of the manuscript.
Any comparison with prior art/(only robot centric/human-centric) to show the improvement in flexibility/adaptability/productivity is suggested
Overall it’s a wonderful framework with experimental justifications.
All the best
Round 2
Reviewer 3 Report
I have had a look at the paper again to check the blue paragraphs. The human-robot collaborative paper is in a good shape now. Reading the response letter, I realized what are the paper contributions in its own.
All issues are resolved in the current version. The paper has already studied the robotic problem, proposed the method accurately, and explained human-aware and user-aware control, etc.
My conclusion is that the paper can be accepted as it is.